# Spatial and Temporal Characteristics and Influencing Factors of G20 Box Office Revenues: A Film Geography Perspective

**Jiejie Wang [1], Mengli Zhang [2], Paul Adams [3], Peng Zheng [2,*] and Xiaoli Wang [2]**

1   School of Tourism and Events Management, Henan University of Economics and Law,
    Zhengzhou 450046, China
2   School of Management, Zhengzhou University, Zhengzhou 450001, China
3   Department of Geography and the Environment, The University of Texas, Austin, TX 78712, USA
*   Correspondence: zhengpeng511@163.com

**Abstract:** The geography of film is an interdisciplinary field of geography and communication. Understanding the spatial and temporal characteristics of the global box office scale represented by the G20 is important for expanding the research scope of film geography focusing on "space" and promoting the development of film industries around the world. This paper studies the spatial and temporal characteristics and influencing factors of G20 box office revenues 2003–2019 using the Theil coefficient, spatial analysis, and a panel vector auto-regressive model. According to the research: (1) the distribution of the top box office revenues within the G20 is obvious; the scale differences among these countries are gradually decreasing and the scale differences between China and the United States are the most significant; (2) the box office differences of the G20 are gradually decreasing; the Asian movie market represented by China and South Korea is developing rapidly; (3) from the perspective of the transfer of movie box office scale types within a Markov Chain, the number of countries in type II and III is the largest and the transfer among different types is mainly to high-level types; and (4) the factors influencing the box office of G20 movies are the number of screens, per capita gross national income, the working-age population, and GDP, respectively.

**Keywords:** spatial and temporal characteristics; influencing factors; box office revenues; film geography; G20





## 1. Introduction

After the 1990s, the "cultural turn" of geography and the "spatial turn" of communication interacted with each other and gradually formed Media Geography, a discipline addressing the relationships and interactions between people and media and between society and geography [1]. On this basis, as a facet of the media geography research field, film geography also has two paradigms focusing on "culture" and "space". The research content of film geography, which focuses on "culture", mainly emphasizes the significance of symbols and uses film texts to analyze the presentation of different subjects to places [2], researchers try to explain and understand the various meanings of cultural texts. To be specific, they pay attention to how films represent places and reveal the power relationships behind the representation [3]. However, some scholars worry that research focusing too much on text and content will gradually separate film geography from the consideration of space and matter, which will cause the emerging film geography tradition to be anemic [4]. Therefore, film geography research that tends to be "spatial" is still a hot spot in the academic field and the existing research in this field is mainly in two aspects. The first aspect is the spatial clustering characteristics of cinemas and shooting bases and the factors influencing their distribution. The spatial aggregation level of cinemas and shooting bases has become one of the main signs of the development level of urban cultural industry [5]. In the field of spatial distribution, it basically focuses on the evolution of urban socio-economic space [6], demographic distribution of the movie-going

market, distance proximity and traffic conditions [7], and location selection [8]. The second aspect is the characteristics and influencing factors of the film industry and box office. The agglomeration level of the film industry and the film box office revenues are important indicators to measure the development status of the film industry [9–11] and they are also important elements to reflect the cultural and economic industries and people's cultural lives [12]. Within countries, the film box office in different cities is affected by factors such as the level of economic development [13], but the very fast development of small- and medium-sized cities causes the monopoly of the top city in box office to not be strong [14]. In the comparison of different countries, some researchers have constructed an evaluation system of world film powerhouses, comparing two broad aspects of domestic influence and international influence [15]; some researchers have analyzed the characteristics of the film industry based on multi-country panel data [16]; some researchers have explained the changes in the film box office in several countries in the East and West from the perspective of cross-cultural differences [17]; and some researchers have analyzed the box office in terms of Internet word-of-mouth [18,19], social media [20], consumption habits [21], etc.

To sum up, there has been more research in the field of film geography, which tends to be "spatial", and this has laid a good foundation for further work. However, most of the existing studies have focused on the box office revenues of individual films [22,23] or films in a particular country [24,25]. Moreover, studies on multiple countries have suffered from endogeneity and underrepresentation [16], without examining the overall pattern of the evolution of the world box office from a more macroscopic perspective of time and space. Therefore, what are the development trends of the box office revenues in countries around the world within the 21st century? What are the influencing factors leading to these changes? All these are urgent questions to be answered in the further development of film geography. Understanding the structure, differences, and evolution of the world film box office in the time dimension represented by the G20 will not only expand the limits of focusing on a certain film or a certain country's film, but also enable us to master the changes of each country's film box office in the world, as well as deeply analyze the influencing factors that shape such changes and provide suggestions for the development of each country's film industry. In view of this, based on the film box office data of the group of twenty (G20) released by the world-famous box office statistics website, Box Office Mojo (https://www.boxofficemojo.com, accessed on 16 October 2021) (Supplementary Materials), from 2003 to 2019, this paper analyzes the hierarchical structure of the box office revenues and the distribution of the box office scale across large-scale regions from the global level, and conducts a linear correlation and panel vector autoregressive analysis on the temporal and spatial characteristics and their influencing factors. Moreover, it adopts a variety of geospatial analysis methods and intuitively reflects the spatial and temporal evolution characteristics of the box office scale of G20 films.

## 2. Materials and Methods

### 2.1. Study Area

The group of twenty (G20) is composed of the group of seven (the United States, the United Kingdom, France, Germany, Italy, Canada, and Japan), the BRICs (China, India, Brazil, Russia, and South Africa), seven important economies (Australia, Mexico, South Korea, Turkey, Indonesia, Saudi Arabia, and Argentina), and the European Union. The member states cover a wide range, encompassing the interests of developed and developing countries as well as the balance of different regions. It is the most representative and influential international economic cooperation forum globally. The G20, accounting for about 90% of the world's GDP and nearly 70% of the world's population, plays an important role in promoting international cultural exchanges and sustainable economic growth and provides a new impetus and opportunities for improving the economic and cultural development of the international community [26].

*2.2. Data Sources*

The paper firstly collected data from professional websites of movie box office statistics in different countries and compared them with Box Office Mojo's movie box office data and found that the data on Box Office Mojo's website were more conducive to reducing the errors caused by different data units, so the paper used Box Office Mojo's movie box office data as a final selection. In addition, the data on other relevant indicators were collected from the official government statistics websites of each country and the official website of UNESCO and the relevant exchange rates were converted according to the average exchange rates of the year. The statistics used here are all from Box Office Mojo, the world's most authoritative film revenue website (https://www.boxofficemojo.com/) [27] the World Bank Database, and the official economic and cultural websites of governments. Among them, Box Office Mojo only covers North American film revenues, so this paper calculates the box office of the United States and Canada in the proportion of 9:1 [21]. In addition, small amounts of missing individual year data are filled using regression analysis through SPSS19.0 software.

It should be noted that the main reasons for not studying the EU and Saudi Arabia in the research are as follows: (1) the EU has 27 member states, including Germany, France, and Italy, and it is difficult to obtain the complete box office data of EU films; (2) Saudi Arabia did not lift the film ban until 2018 [28], so no data on movie box office receipts can be found. Furthermore, the primary reasons for choosing 2003–2019 as the research time span are: (1) China carried out a number of reforms in the film industry in 2002, including the promulgation of the regulations on the administration of films, the reform of the cinema system throughout the country, and measures breaking the monopoly of a single state provider over the import and distribution of foreign movies; (2) from the macro perspective of world films, the world film box office in 2003 was at the beginning of the rebound from a low point [29]; and (3) affected by COVID-19, the data in 2020 are not included in the paper.

*2.3. Research Method*

Based on the film box office revenue data of G20 countries from 2003 to 2019, this paper adopts the rank-size rule and primacy index to judge the relationship between the film box office of different countries and their corresponding rank in the whole system and depicts the scale distribution of the G20 film box office. Secondly, the temporal and spatial characteristics of the G20 film box office scale in general, within and among regions, are analyzed using the Theil coefficient. Furthermore, ArcGIS 10.2 software, the center of gravity transfer, Markov transfer matrix and other methods are used to calculate the center of gravity trajectory evolution and the type of transformation process of the film box office in G20 countries. Finally, the PVAR model is established to analyze the influencing factors of the film box office.

## 3. Temporal and Spatial Pattern Analysis

*3.1. Box Office Scale and Spatial and Temporal Characteristics*

This paper uses the ratio of the box office of G20 countries to the average box office of all countries in the world as the film box office scale index of all countries. The film box office scale index of the United States decreased from 8.5139 to 5.6603, showing a declining trend year by year. The box office scale index of Chinese films increased from 0.0883 to 4.2186, showing a yearly increasing trend, and the gap between China and the United States gradually decreased. The film box office indexes of other G20 countries had little change, with the change range below 1. Among them, while the box office indexes of Japan, Germany, and some countries decreased, the indexes of Russia, South Korea, and other countries increased slightly (Table 1).

**Table 1.** Box office scale index of G20 films.

| Nation | 2003 | 2019 | Change Range | Direction |
|---|---|---|---|---|
| China | 0.0883 | 4.2187 | 4.1304 | + |
| United States of America | 8.5139 | 5.6603 | 2.8536 | − |
| Japan | 1.8095 | 1.2675 | 0.542 | − |
| Germany | 0.9197 | 0.4464 | 0.4733 | − |
| Russia | 0.079 | 0.4349 | 0.3559 | + |
| South Korea | 0.4502 | 0.7784 | 0.3282 | + |
| Canada | 0.946 | 0.6289 | 0.3171 | − |
| France | 0.9948 | 0.6838 | 0.311 | − |
| Australia | 0.5606 | 0.3525 | 0.2081 | − |
| Italy | 0.4892 | 0.2879 | 0.2013 | − |
| Britain | 1.2503 | 1.0506 | 0.1997 | − |
| Brazil | 0.1673 | 0.314 | 0.1467 | + |
| India | 0.8773 | 1.0103 | 0.133 | + |
| Indonesia | 0.2385 | 0.2962 | 0.0577 | + |
| South Africa | 0.0721 | 0.0448 | 0.0273 | − |
| Mexico | 0.408 | 0.3905 | 0.0175 | − |
| Argentina | 0.0591 | 0.056 | 0.0031 | − |
| Turkey | 0.0763 | 0.0781 | 0.0018 | + |

Judging from the overall scale, the G20 film box office tripled in the past 17 years. At the same time, the standard deviation decreased from 1.8836 to 1.4526, indicating that the dispersion of the film box office in various countries was decreasing. By using a primacy index to measure the concentration of film box office distribution, it is found that the primacy index of the G20 film box office shows a downward trend year by year, from 4.705 in 2003 to 1.3417 in 2019, with the differences in film box office size between the first-order and second-order countries gradually narrowing. In order to better explain the distribution of the film box office scale in G20 countries, the rank-size rule [29] was used for further analysis. The results showed that the goodness of fit R2 of the ranking scale of G20 film ticket revenue from 2003 to 2019 was basically above 0.75, reflecting the applicability of the power function to the distribution of the ranking scale of the film box office. As can be seen from the scale distribution of the G20 film box office (Figure 1), the higher the national ranking, the faster the growth of the box office scale. According to the double logarithm regression analysis, the concentration index of G20 film box office fluctuates and decreases gradually as a whole, but the concentration index is always above 1. On the one hand, it shows that the film box office of high-order countries still has outstanding advantages, while the film box office of low-order countries is relatively backward with significant polarization characteristics. On the other hand, it further indicates that the monopoly degree of the G20 film box office scale has decreased on the whole and that the relative difference degree has decreased, which shows that the agglomeration degree of the G20 film box office was decreasing (Table 2).

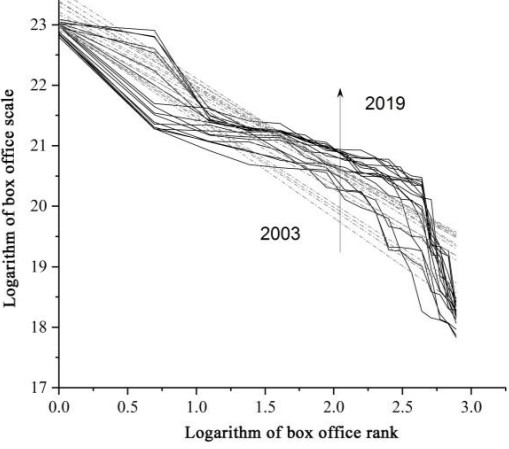

**Figure 1.** Logarithmic curve of rank scale of box office distribution of G20 films from 2003 to 2019.

**Table 2.** Box office scale of G20 films.

| Year | Overall Scale | Mean Value | Standard Deviation | Primacy Index | Film Market Size | $R^2$ |
|------|--------------|-----------|--------------------|---------------|-----------------|-------|
| 2003 | 17,439,979,922 | 968,887,773 | 1.8836 | 4.705 | lny = 22.9896 − 1.5905 lnx | 0.8845 |
| 2004 | 18,746,964,576 | 1,041,498,032 | 1.7609 | 4.2525 | lny = 22.9631 − 1.4901 lnx | 0.9213 |
| 2005 | 18,455,605,713 | 1,025,311,428 | 1.8160 | 4.688 | lny = 22.9866 − 1.5265 lnx | 0.8966 |
| 2006 | 18,979,089,401 | 1,054,393,856 | 1.6575 | 4.5583 | lny = 22.9787 − 1.4686 lnx | 0.8634 |
| 2007 | 21,302,980,578 | 1,183,498,921 | 1.5221 | 4.7631 | lny = 22.9107 − 1.3144 lnx | 0.8392 |
| 2008 | 22,981,557,432 | 1,276,753,191 | 1.4755 | 4.6108 | lny = 22.9981 − 1.3185 lnx | 0.7962 |
| 2009 | 22,937,966,528 | 1,274,331,474 | 1.4755 | 3.9357 | lny = 22.9988 − 1.3206 lnx | 0.7983 |
| 2010 | 26,043,075,248 | 1,446,837,514 | 1.4158 | 3.7902 | lny = 22.9609 − 1.2018 lnx | 0.7666 |
| 2011 | 27,280,009,498 | 1,515,556,083 | 1.3425 | 4.1887 | lny = 23.0509 − 1.2230 lnx | 0.7335 |
| 2012 | 28,358,151,647 | 1,575,452,869 | 1.3334 | 3.6829 | lny = 23.0972 − 1.2188 lnx | 0.7923 |
| 2013 | 29,162,601,796 | 1,620,144,544 | 1.3243 | 2.734 | lny = 23.1703 − 1.2528 lnx | 0.7632 |
| 2014 | 29,724,398,111 | 1,651,355,451 | 1.3388 | 2.1645 | lny = 23.2154 − 1.2716 lnx | 0.8132 |
| 2015 | 29,748,115,946 | 1,652,673,108 | 1.4154 | 1.4206 | lny = 23.3922 − 1.4156 lnx | 0.8691 |
| 2016 | 30,971,016,633 | 1,720,612,035 | 1.3932 | 1.6362 | lny = 23.3120 − 1.3253 lnx | 0.8472 |
| 2017 | 33,191,972,855 | 1,843,998,492 | 1.4414 | 1.2749 | lny = 23.3740 − 1.3366 lnx | 0.8698 |
| 2018 | 33,349,596,112 | 1,852,755,340 | 1.4799 | 1.1196 | lny = 23.4767 − 1.4198 lnx | 0.8535 |
| 2019 | 34,027,627,045 | 1,890,423,725 | 1.4526 | 1.3417 | lny = 23.5764 − 1.4644 lnx | 0.8457 |

### 3.2. Spatial and Temporal Characteristics of Box Office Scale

This paper adopts a Theil coefficient, which is better than Gini coefficient in domain decomposition, to analyze the spatial differences among and within the G20 film box office regions.

$$T_p = \sum i \sum j \frac{Y_{ij}}{Y} \log \frac{Y_{ij}/Y}{N_{ij}/N} = \sum i \frac{Y_i}{Y} T_{pi} + T_{br} = T_{wr} + T_{br} \tag{1}$$

$Y_{ij}$ and $N_{ij}$ are the film box office revenues and population of *I* area *j* nation, respectively; $Y_i$ and $N_i$ are film box office revenue and population of *I* area, respectively; *Y* and *N* are total box office size and population of G20, respectively. The box office difference (*Tp*) of G20 films can be decomposed into the sum of intra-regional and inter-regional differences.

From 2003 to 2019, the overall Theil coefficient, interregional Theil coefficient, and intraregional Theil coefficient of G20 film box office with a fluctuating downward trend indicate that the overall, inter-regional, and intra-regional box office differences of G20 were decreasing year by year. It can be further inferred that the competition in the G20 film box office market and the prosperity of the global film box office market became fiercer. To some extent, the occurrence of such changes indicates both economic globalization and cultural globalization [30]. Meanwhile, the contribution rate of the Theil coefficient within each region increased steadily and the contribution rate of between-region differences to the Theil coefficient gradually decreased. The regional differences have become the dominant reason affecting the overall differences in recent years.

Further comparisons of the Theil coefficient were performed in the four regions of North America, Europe, Asia, and other continents from 2003 to 2019 (Figure 2). The variation range of the Theil coefficient in North America, Europe, and other continents remained relatively stable, with little differences in the box office scale. The Theil coefficient in Asia changed the most, from 0.5691 in 2003 to 0.1635 in 2019. With the rise of Asian films represented by China and South Korea, the whole Asian film market developed rapidly and the differences in the box office scale among Asian countries gradually decreased.

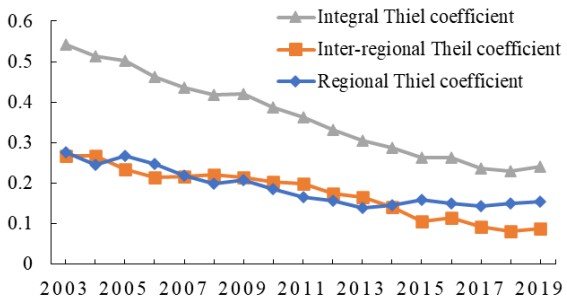 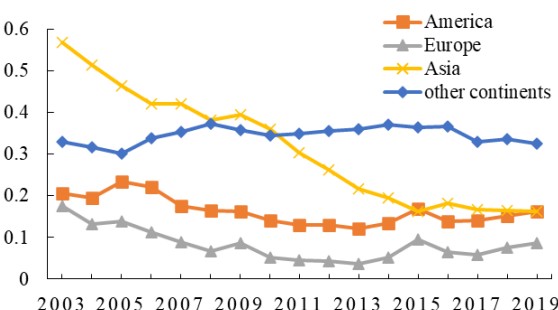

**Figure 2.** Decomposition of the overall, interregional, and intraregional box office differences of G20 films. Note: In the (**left**) of the figure, the gray triangle represents the fluctuation of the overall Tyr coefficient value, the orange square represents the fluctuation of the inter-regional Tyr coefficient value, and the blue diamond represents the fluctuation of the intra-regional Tyr coefficient value; in the (**right**) of the figure, the orange square represents the fluctuation of the Tyr coefficient value in the American region, the gray triangle represents the fluctuation of the Tyr coefficient value in the European region, the yellow double arrow represents the fluctuation of the Tyr coefficient value in the Asian region, and the blue diamond represents the fluctuation of the regional Tyr coefficient in other states.

### 3.3. Temporal and Spatial Evolution of Box Office Scale

3.3.1. Center of Gravity Distribution Trajectory Evolution

The center of gravity for box office revenue captures the geographical center of the revenue distribution and its spatial aggregation. The geographical transfer of the center of gravity follows the track created by movements in the center of gravity of the G20 film box office and also reflects changes in the direction of film box office revenue flows [31].

$$G(x,y) = \frac{P_i \cdot Q(x_i, y_i)}{\sum P_i} \tag{2}$$

*G* is the center of gravity of the film scale, *Q* represents the center of the administrative region, *p* represents the film scale, *I* represents the count of the administrative region, *X* and *Y* represent the longitude and latitude of the gravity center, respectively, and *X* and $Y_i$ represent the longitude and latitude of the administrative region, respectively.

The film box office from 2003 to 2019 was selected as the spatial element of the growth scale of the G20 film box office. In terms of the G20 film box office, it can be seen from Figure 3 that the center of gravity from 2003 to 2019 showed a track moving substantially from west to east and slightly from north to south, with obvious spatial diffusion. Among them, the minimum displacement of the G20 film box office scale was from 2012 to 2013, with the gravity center moving 70.5342 km to the southeast. The maximum displacement is from 2014 to 2015 when the gravity center moved 661.8278 km to the east (Table 3). Although the box office scale of North American films plays a dominant role in the G20, the box office scale of East Asian countries represented by China and South Korea increased rapidly during the research period, resulting in a new space for the overall eastward focus track.

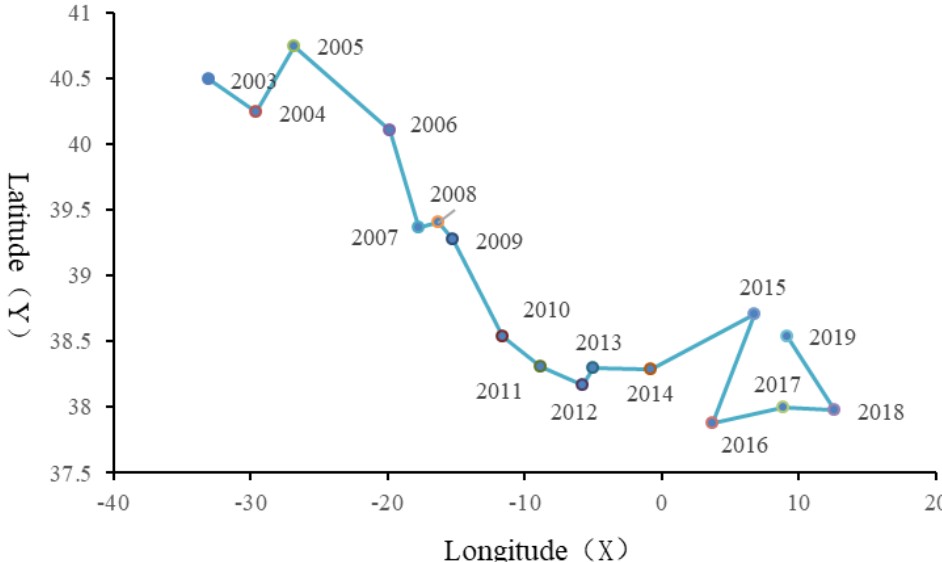

**Figure 3.** The track of the center-of-gravity transfer of box office scale of G20 films.

**Table 3.** Box office scale transfer of G20 films.

| Year | Displacement (km) | Year | Displacement (km) |
|---|---|---|---|
| 2003–2004 | 296.9053 | 2011–2012 | 269.9283 |
| 2004–2005 | 241.2293 | 2012–2013 | 70.5342 |
| 2005–2006 | 598.8595 | 2013–2014 | 366.78 |
| 2006–2007 | 191.8569 | 2014–2015 | 661.8278 |
| 2007–2008 | 122.7804 | 2015–2016 | 283.1328 |
| 2008–2009 | 95.3159 | 2016–2017 | 451.1786 |
| 2009–2010 | 329.4448 | 2017–2018 | 329.2107 |
| 2010–2011 | 239.691 | 2018–2019 | 309.5141 |

### 3.3.2. Markov Transition Probability Evolution

The Markov chain is a discrete random process, which is used to study the dynamic evolution process of G20 film box office size distribution and form a K × K-state transition matrix [32].

$$M_{ij} = n_{ij}/n_j \tag{3}$$

Using a Markov transfer matrix for calculation, the process is as follows: ① The Euclidean distance clustering method of spss software is used to Q-cluster the G20 film box office each year and the G20 film box office revenues are divided into four types from low to high as I, II, III, and IV, respectively; ② Constructing a probability matrix of shifts in the G20 film box office between four types from 2003–2019 to obtain a 4 × 4 Markov shift probability matrix, it can be seen from Table 4 that the higher the frequency and probability value on the diagonal, the greater the possibility of maintaining the same type in the next year. There is no cross-level transfer among different box office types in various countries and the transfer trend among different types is mainly to high-level types; ③ The evolutionary trend of the G20 film box office revenues is studied by calculating the frequency and transfer probability of transfer between different types. The concentration degree of transfer probability among different types is IV > I > III > II, indicating that the type of transfer of countries with high film box office revenues is the main reason for the change.

**Table 4.** Markov transition matrix of different countries in different box office scale types (2003–2019).

| $t_i/t_{i+1}$ | n | Transfer Frequency | | | | Transition Probability | | | |
|---|---|---|---|---|---|---|---|---|---|
| | | I | II | III | IV | I | II | III | IV |
| I | 85 | 78 | 7 | 0 | 0 | 0.9176 | 0.0824 | 0 | 0 |
| II | 95 | 2 | 84 | 9 | 0 | 0.0211 | 0.8842 | 0.0947 | 0 |
| III | 92 | 0 | 4 | 84 | 4 | 0 | 0.04348 | 0.9130 | 0.0435 |
| IV | 34 | 0 | 0 | 2 | 32 | 0 | 0 | 0.0588 | 0.9412 |

According to the visual analysis of the spatial distribution of G20 film box office scale (Figure 4), in 2003, only the United States belonged to type IV, Britain and Japan belonged to type III, and the other countries were distributed in type II and type I. In 2011, China and Russia changed greatly from type I countries to type III countries. In 2019, the United States, Japan, and China became type IV countries; only South Africa, Turkey, and Argentina were type I countries; Germany, Italy, Brazil, and other countries were type II countries; and Britain, France, Canada, India, and South Korea were type III countries. It can be seen that the transfer of film box office types in most countries achieved a transition, in which China completed a three-level jump from type I to type IV.

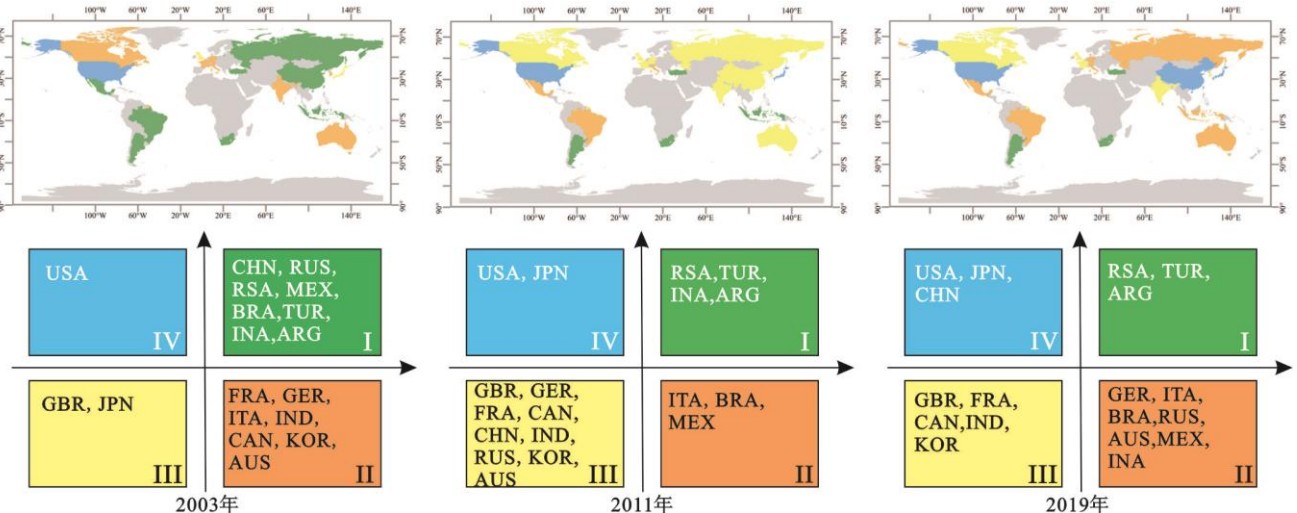

**Figure 4.** Spatial distribution of box office scale types of G20 films in 2003, 2011, and 2019. Note: the drawing is based on the standard map with the drawing approval No. GS (2016) 1667 downloaded from the standard map service website of the National Bureau of surveying, mapping, and geographic information and the base map is not modified.

## 4. Analysis of Influencing Factors

### 4.1. Selection of Influencing Factors

Building on the analysis of the temporal and spatial pattern of the G20 film box office revenues, we next explore its influencing factors. The factors influencing the overall box office of a country are different from those influencing the factors of a particular film [33]. The national box office revenues are an important reflection of the country's film industry and socio-economic development and they are closely related to the movie-going population and their purchasing power at the demand level and to the facility base of the film industry at the supply level [34]. Considering the universality of application in different countries and the convenience of data access, the paper draws on the existing research results [35]; it is possible to further analyze the influencing factors of G20 film box office scale relative to four aspects: economic development, market development, film industry, and purchasing power. Moreover, gross domestic product (GDP) is used to measure the economic development level of the country, the working population is used to

represent the potential of market development and the advantages of human resources, the number of screens is used to represent the development degree of the film industry, and the per capita gross national income (GNI) is used to measure the purchasing power of residents. The GDP, working age population, number of screens, and GNI of G20 countries from 2003 to 2019 were selected for linear correlation analysis with box office revenues (Table 5). The results showed that the correlation coefficients of the four influencing factors were 0.893 ($p < 0.01$), 0.369 ($p < 0.01$), 0.849 ($p < 0.01$), and 0.452 ($p < 0.01$), respectively, and the positive correlation of all influencing factors was significant.

**Table 5.** Correlation analysis of box office revenues with influencing factors in G20 countries.

| Influencing Factor | Specific Indicator | Correlation Coefficient | z | p |
|---|---|---|---|---|
| Economic development level | GDP | 0.893 *** | 20.72 | 0.000 |
| Market development degree | Population of working age | 0.369 *** | 4.21 | 0.000 |
| Film industry factors | Number of screens | 0.849 *** | 20.05 | 0.000 |
| Residents' purchasing power | GNI | 0.437 *** | 4.97 | 0.000 |

*** means the Level of Significance when $p < 0.01$.

### 4.2. Empirical Analysis of Influencing Factors

After the selection of the above influencing factors, we construct a panel vector autoregressive model with the GDP, working age population, number of screens, GNI, and G20 film box office panel data of 18 countries from 2003 to 2019 to explore the dynamic relationships and causal mechanisms among the G20 film ticket office as well as the influencing factors in time and space.

#### 4.2.1. Establishment of PVAR Model

The basic expression of PVAR model is as follows:

$$Y_{it} = \alpha_0 + \sum_{j=1}^{p} \alpha_j Y_{t,t-j} + \sum_{j=1}^{p} \beta_j X_{j,t-j} + \theta_i + \varphi_t + \varepsilon_{it} \tag{4}$$

$Y_{it}$ indicates the box office of G20 films; $I$ and $t$ represent the cross-section and time series of countries, respectively; $\alpha_0$ is an intercept item; $\alpha_j$ and $\beta_j$ indicates the coefficient to be estimated of the dependent variable; $p$ represents the lag order; $X_{j,t-j}$ explanatory variables represents the box office of G20 films; and $\theta_i$, $\varphi_t$, and $\varepsilon_{it}$ represent individual and time fixed effects and white noise disturbance terms, respectively. In order to avoid the heteroscedasticity problem of model estimation results, each variable is logarithmicized, namely *lngdp, lnworking, lnscreen,* and *lngni*.

#### 4.2.2. Stationary Test

The stationarity test is an important prerequisite for constructing a PVAR model. Since the model setting includes time series, the unstable data will lead to pseudo regression in the estimation results of the model, which cannot truly reflect the internal logical relationship between various variables. In this paper, Stata 15.0 software is used to conduct *IPS* (applicable to heterogeneous unit root hypothesis) and *LLC* (applicable to homogeneous unit root hypothesis) for each variable and investigate the stationarity of five variables, respectively (Table 6). It is found that some variables accept the original assumption as non-stationary data, but all the data reject the original assumption as stationary data after the first-order difference of variables, which meets the preconditions for building a PVAR model.

**Table 6.** Unit root test of variables.

| Sequence Name | Inspection Method | Value |
|---|---|---|
| *lnboxoffice* | IPS | −3.4194 *** (0.0003) |
| | LLC | −7.6723 *** (0.000) |
| *lngdp* | IPS | −0.8644 (0.1939) |
| | LLC | −3.5446 *** (0.0002) |
| *lnworking* | IPS | 6.1944 (1.0000) |
| | LLC | 4.5979 (1.0000) |
| *lnscreen* | IPS | −2.6630 *** (0.0039) |
| | LLC | −7.2361 *** (0.000) |
| *lngni* | IPS | −2.2548(0.0121) |
| | LLC | −6.7934 *** (0.000) |
| *dlnboxoffice* | IPS | −13.1305 *** (0.000) |
| | LLC | −13.1478 *** (0.000) |
| *dlngdp* | IPS | −7.2742 *** (0.000) |
| | LLC | −10.1554 *** (0.000) |
| *dlnworking* | IPS | −8.7959 *** (0.000) |
| | LLC | −10.6575 *** (0.000) |
| *dlnscreen* | IPS | −10.2017 *** (0.000) |
| | LLC | −10.3585 *** (0.000) |
| *dlngni* | IPS | −3.7723 *** (0.0001) |
| | LLC | −6.2722 *** (0.000) |

Note: the inspection value of $p$ is indicated in brackets, *** indicates significant at the 1% level.

### 4.2.3. Lag Order Selection

Before using the PVAR model for estimation, the optimal lag order of the model shall be determined. Based on the existing PVAR2 program research results [36], AIC, BIC, and hqic statistics are constructed to determine the optimal lag order. Then, referring to the research of relevant scholars [37], the standard of the best lag order is: under the AIC, BIC, and hqic criteria, the minimum value. In order to avoid the loss of degrees of freedom caused by the lag order, the smaller lag order should be as far as possible. On the basis of the above principles and the test results of AIC, BIC, and HQIC criteria (Table 7), the optimal lag order is selected as 1.

**Table 7.** AIC, BIC, and HQIC criteria test.

| Lag Order | AIC | BIC | HQIC |
|---|---|---|---|
| 1 | −13.968 * | −12.4354 * | −13.3526 * |
| 2 | −12.9458 | −10.985 | −12.1568 |
| 3 | −12.953 | −10.5166 | −11.9707 |
| 4 | −12.4908 | −9.52185 | −11.2914 |

Note: * indicates the best lag time of AIC, BIC, and HQIC test results.

### 4.3. Impulse Response Analysis

The impulse response analysis is to investigate the dynamic influence trend of a positive standard deviation of dependent variables on the film box office and can reflect the dynamic influence track and development trend of each variable on the film box office. Figure 5 shows the impulse response diagram of each variable with a lag of 10 periods after 200 Monte Carlo simulations.

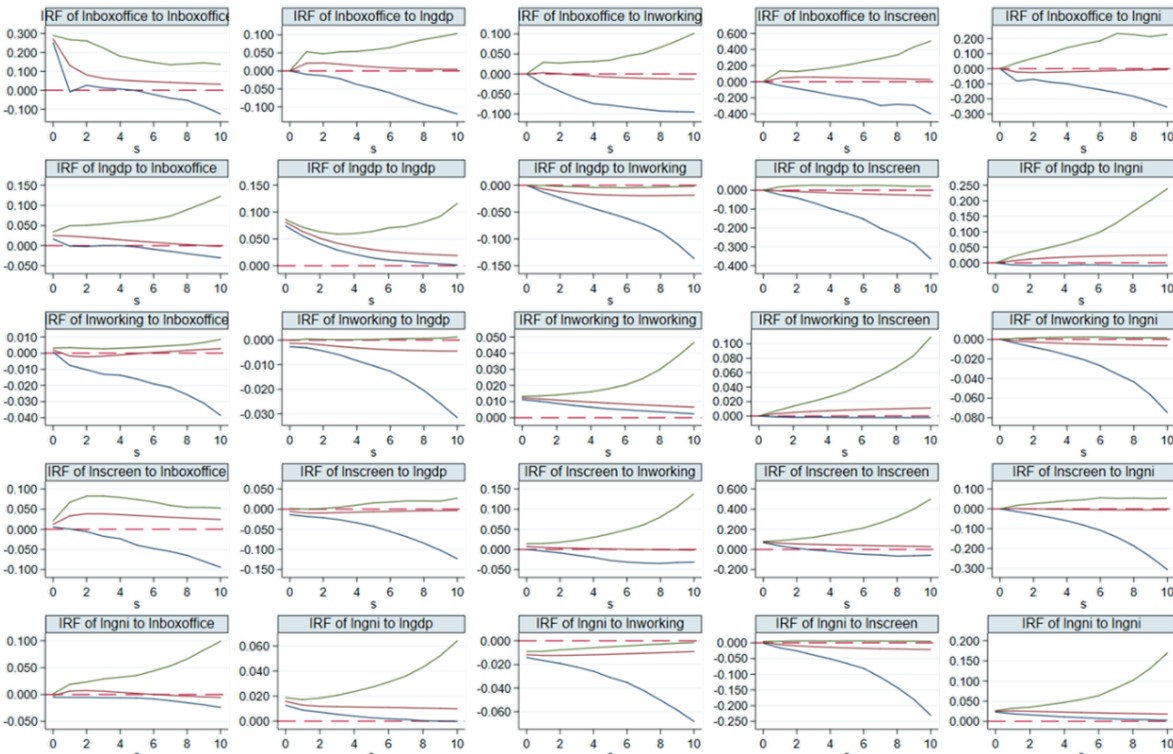

**Figure 5.** The impulse effect of the box office of G20 countries on the impact of various influencing factors. Note: The horizontal axis indicates the number of lags in the effect of the shock, the vertical axis indicates the change in the explanatory variable, the red line in the middle indicates the impulse response function, and the blue and green lines on either side indicate the positive and negative two times standard deviation bands. The red line, indicates how the explanatory variable changes after giving a shock to the explanatory variable and the effect of the shock on a variable at different periods.

As shown in Figure 5, from the impulse impact of GDP on the film box office, it can be seen that the impulse response produces a positive effect at the initial stage, but then the positive effect gradually decreases. The working age population has a positive effect on the impulse impact of the film box office at the initial stage, but then begins to decline, which shows that the working age population has a certain role in promoting the film box office. From the impulse impact of the number of screens on box office revenues, it can be concluded that the impulse response value has a positive effect in the initial stage, then begins to rise, and reaches the maximum value in the second stage, but the maximum value fails to exceed 0.1 and gradually falls to zero, indicating that the number of screens plays an obvious role in promoting the film box office. The impulse response of GNI to the film box office is zero at the beginning, then fluctuates and increases, resulting in a positive effect, in which the change is not obvious, and finally it arrives at zero.

On the whole, the impulse response of the film box office to GDP, the working age population, the number of screens, and GNI do not respond at the initial stage. Later, the response value changes, the fluctuation range is small, and it finally disappears. The response value of the film box office to GDP first increases and then decreases, reaching its maximum in the first phase. The impulse response impact of the film box office as a result of the working age population and the number of screens is small and fluctuates around zero. The response value of film box office to GNI first decreases and then increases, reaching the minimum in the first period.

*4.4. Variance Decomposition*

The variance decomposition examines the proportion of the unit impact on the box office revenues from each variable, which can measure the impact of each variable in

different periods. Therefore, this paper presents the variance decomposition results of film box office revenue, GDP, the working age population, number of screens, and the per capita GNI in periods 1, 20, 50, and 70 (Table 8).

**Table 8.** Panel error variance decomposition.

| Variable | Period | dlnboxoffice | dlngdp | dlnworking | dlnscreen | dlngni |
|---|---|---|---|---|---|---|
| dlnboxoffice | 1 | 1.000 | 0.000 | 0.000 | 0.000 | 0.000 |
| | 20 | 0.803 | 0.014 | 0.016 | 0.146 | 0.021 |
| | 50 | 0.761 | 0.018 | 0.025 | 0.165 | 0.032 |
| | 70 | 0.754 | 0.018 | 0.024 | 0.170 | 0.033 |

From the 70th period, the fluctuation trend of each variable begins to stabilize. Therefore, based on the results of variance decomposition in period 70, the interaction between the film box office and influencing factors is analyzed. In the variance decomposition of G20 film box office scale, its own development contributes to 75.4% of the explanatory power, which has the characteristics of sustainability. In addition, the contribution rate of the number of screens to the film box office is the highest, reaching 17%, while the contribution rates of GDP, working age population, and GNI to the film box office scale of various countries are only 1.8%, 2.4%, and 3.3%, respectively. It can be seen that the contribution of various influencing factors to the box office of G20 films ranks as: number of screens > GNI > number of working age population > GDP.

## 5. Conclusions and Discussion

### 5.1. Discussion

Although the proportion of box office revenues to the total national economy is not high [21], the strong cultural influence of the film industry is an issue that cannot be ignored [38]. If the performance of this cultural influence is limited to the analysis of the content of individual film texts without considering the overall change in the box office scale of films among countries, the importance of the cultural influence may be underestimated [39]. This paper examines the evolution of the spatial and temporal characteristics and the influencing factors of G20 film box office revenues from a global perspective, which not only strengthens the interdisciplinary research of geography and communication, but also widens the research scope of film geography.

Throughout the development process of the spatial and temporal characteristics of the film box office scale of G20 countries in the past 17 years of this century, the global film industry is booming, which is manifested in the evolution trend of globalization and diversification. The leading position of the American film box office scale has not been shaken, but the emerging film industry represented by China, South Korea, and Russia is rising rapidly, which is not only an important engine of global film box office growth, but also changes the space–time pattern of global film box office scale. As for the influencing factors of the film box office scale, this paper is in accordance with the previous results [16], which insist that the GDP, the working age population, and the per capita gross national income can promote the increase in film box office scale, but the contribution is not prominent. As for the number of screens, we find that this variable contributes greatly to the box office revenues, which is different from the view of Shania and Pu Yongjian (2012) that the number of screens plays a limited role in expanding the scale of the film box office [16]. Using the Chinese film market as an example, the number of Chinese screens surged after 2010 and surpassed the United States in 2016. The most obvious change is the increasing number of screens in small- and medium-sized cities in the third and fourth tier and "small town youth" has become a new force in the growth of the film box office [40]. Although China's major film consumption centers are still in first and second tier cities, the growth rate of the box office scale leads in third and fourth tier cities. This reflects the characteristics of films as medium- and low-end mass cultural products. The vast

number of small- and medium-sized cities in China are in the initial stage of economic development and cultural consumption and the box office of films is growing rapidly. In economically developed cities, the film market tends to be saturated, the box office growth slows down gradually, and more consumers begin to pay attention to high-level cultural products such as concerts and art exhibitions [41]. In addition, "the role played by government planners" is also one of the important reasons for the prosperity of China's film market (http://fashion.people.com.cn/n1/2019/1014/c1014-31398141.html?utm_source=ufqinews, accessed on 14 October 2021). In order to promote the rapid and balanced growth of the film market, the Chinese government has gradually liberalized the access policy of the film threshold since 2003, established the status of non-public ownership in the film industry, and provided a policy guarantee for this. Xu Zhang and Yajuan Li (2018) argue that the film industry in China have been affected both by market and non-market factors, compared to its counterparts in Western countries, the Chinese state tends to play a more influential role in shaping the patterns of film production and consumption in the country [41]. Using South Korea as another example, despite having a population of only 50 million, the national audience grew from 150 million to 224 million between 2010 and 2019, which is largely attributed to the success of local Korean films, as illustrated by the success of the movie *Parasite*. This is due not only to the government's focus on funding the film industry, but also to the establishment of the government-backed semi-private and semi-official Korean Film Promotion Committee. Additionally, unlike the U.S., South Korea allows for vertical integration between distributors and exhibitors [42]. This has contributed to South Korea's ability to resist pressure from the Motion Picture Export Association to adhere to a certain percentage of screen quota systems for domestic film screening days, so South Korea has an incentive to build more movie screens. Of course, China, South Korea, and Russia still have much room for improvement compared to the United States, France, and Japan in terms of the industry environment represented by film technology and film education and industry competitiveness represented by the overseas film market share and international mainstream film awards [43].

### 5.2. Conclusions

#### 5.2.1. Academic Implications

Geography and communication have never been separated from each other's influence. As David Harvey says, "Geography is too important to be studied by geographers alone," and communication is so complex that it can never be exhausted by communication scholars [44]. This paper investigates the spatial characteristics of the G20 box office scale and the factors influencing it from the perspective of the "spatial" tendency of film geography through various geospatial analysis methods. Although the United States, as the country ranking first in the world, still occupies the absolute dominant position, the gap between the box office scale of each country is narrowing year by year, among which the Asian countries represented by China and South Korea are developing rapidly. In terms of box office scale differences among G20 countries, overall, inter-regional, and intra-regional differences are decreasing year by year, with intra-regional differences being the main cause of overall differences. In terms of intra-regional variation, intra-regional variation in Asia has been decreasing and changing significantly, while intra-regional variation in the Americas, Europe, and other continents has not changed significantly. Although North America dominates the G20 box office, the rapidly growing box office size in China and South Korea has led to an overall shift in the trajectory of the center of gravity to the east. In terms of the shift in box office scale genres across countries, the shift between different genres is dominated by the shift to higher-level genres and the genre shift in countries with higher movie box office scales is the main reason for the genre shift, with China in particular being the most prominent. In addition, most of the literature focuses on the reasons for the success or otherwise of a particular film at the box office [45], often ignoring the overall pattern of a country's film box office in the world and the factors that influence it. Film is not an isolated industry, but is developed in a certain socio-economic

context and supported by a certain industrial foundation [46], such as film industry policies, film infrastructure, national economic conditions, employment and income of the main consumer groups, education level of the population, etc. These factors determine to a considerable extent the supply, demand, and scale of the film industry. This paper finds that the level of economic development, the degree of market development, film industry factors, and residents' purchasing power are all correlated with the scale of the movie box office in each country. However, the degree of contribution of each factor to G20 movie box office varies significantly and the contribution of each factor is in the order of number of screens, per capita gross national income, the working-age population, and the GDP.

### 5.2.2. Practical Inspirations

This paper explores the evolution of the spatial and temporal characteristics of the box office of G20 countries from a global perspective, which provides data support for each country to understand the position of its film industry scale in the world and its development trend. From the results of the study, with the rapid growth of East Asian countries, represented by China and South Korea, in the scale of movie box office, the primacy ratio of the United States has significantly decreased. However, it does not indicate the decline of the U.S. in the global film market; its absolute dominance in terms of the breadth and depth of film culture influence remains. Referring to the top-grossing movies of the world in 2019, almost all the top-grossing movies in different countries of the world, except for China, South Korea, and Japan, came from the United States. From the perspective of sustainable development of the global film industry, the world box office revenues should show a balanced characterization of globalization and diversification. Only the common progress of the film industry in multiple countries can lead to wider cultural exchanges and mutual appreciation of civilizations worldwide. Regarding the box office influencing factors of the national film industries, the level of economic development within countries and the demand of people's cultural life are important factors affecting the overall box office revenues. For the G20, the rapid development of emerging economies such as China, India, and Brazil provides a huge local market for film industry development. The number of screens, a key factor at the supply level, has become a gas pedal for the development of the movie box office. In the broader developing countries, raising the economic disposable income of all people, actively increasing the number of screens, enhancing the scale of the movie box office, and developing cultural and creative industries led by movies can enhance the national cultural soft power.

### 5.2.3. Future Prospects

Film geography is an emerging research field that integrates geography and communication. Although it has not yet developed into a mainstream branch discipline, an unformed research field can better adjust to the participation and contention among different disciplines and different schools and produce unexpected collisions. As for the research content of film geography, whether it is biased towards "culture" or "space", the distinction between the two is temporary. The integration and embedding of findings from each approach in the other can enable it to be possible to touch on the essence of deep problems. Therefore, in the future, the spatial pattern of film trade imports and exports among G20 countries can be further studied. The comparison between local film box offices and the international film box office is a reflection of the power–influence relationship of culture and politics between countries.

**Supplementary Materials:** Supporting information can be downloaded at: https://www.boxofficemojo. com (accessed on 16 October 2021), the World Bank Database and the official economic and cultural websites of governments.

**Author Contributions:** Conceptualization, P.Z. and J.W.; methodology, P.Z. and M.Z.; software, M.Z. and J.W.; validation, J.W., P.Z., M.Z., P.A. and X.W.; formal analysis, J.W., P.Z. and P.A.; investigation, M.Z.; resources, M.Z.; data curation, J.W. and M.Z.; writing—original draft preparation, P.Z., M.Z.

and P.A.; writing—review and editing, J.W., P.A. and X.W.; visualization, M.Z. and J.W.; supervision, P.A. and J.W.; project administration, J.W. and X.W.; funding acquisition, J.W. and P.Z. All authors have read and agreed to the published version of the manuscript.

**Funding:** This research was supported by the National Social Science Foundation of China (to Peng Zheng, Grant No. 21BGL260); National Natural Science Foundation of China (to Jiejie Wang, Grant No. 41901178); The Ministry of Education of Humanities and Social Science Project of China (to Jiejie Wang, Grant No. 19YJCZH158); Training Program for Young Backbone of Higher Education Institutions in Henan Province (to Jiejie Wang, Grant No. 2020GGJS115); Henan Provincial Higher School Philosophy Social Science Innovation Talent Support Program (to Jiejie Wang, Grant No. 2022-CXRC-26).

**Institutional Review Board Statement:** Not applicable.

**Informed Consent Statement:** Not applicable.

**Data Availability Statement:** Not applicable.

**Conflicts of Interest:** The authors declare no conflict of interest. The funders had no role in the design of the study; in the collection, analyses, or interpretation of data; in the writing of the manuscript; or in the decision to publish the results.

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
