# Peer review of "Spatial and Temporal Characteristics and Influencing Factors of G20 Box Office Revenues: A Film Geography Perspective"

_sustainability, doi:10.3390/su142416535_

Round 1

Reviewer 1 Report

The paper is concerned with investigating the relationship between the film box office of G20 countries, and depicts the scale distribution of film box office based on the data related to film box office revenue in years 2003-2019. The paper consists of five sections: introduction, materials and methods, temporal and spatial pattern analysis, analysis of influencing factors, conclusion and discussion. In my opinion, the paper should be improved in the following directions:

1. The authors should clearly indicate the aspect in which their research affects sustainability. At the moment, there is hard to identify in which aspect their paper is related to journal topics.

2. Section “discussionshould be placed before section “conclusion”. The authors should present the contribution of their research to the theory and practice in a descriptive way (not as a list of four paragraphs) .

3. The abstract could be rewritten towards avoiding unnecessary details that seem to suit better to sections related to data analysis (e.g., “the box office scale index from 0.0883 to 4.2186”; “from 8.5139 to 5.6603”;with the variation range of Theil coefficient 20 decreasing from 0.5691 to 0.1635”). Moreover, the symbol “>” in the last sentence of the abstract is ambiguous.   

4. The authors have listed 47 references at the end of the paper, however, many of these references are not cited in the paper, for example, [4], [7-8], [10-11], [14], [16], [18-21], [26], [34], [37], [41], [43].

Reviewer 2 Report

Reviewer Report for the Manuscript: Sustainability- 2041210

Influencing Factors and Spatial Distribution of
Movie Box Office in the G20

Journal Name: Journal of Sustainability

REVIEWER REPORT

Summary

The article focuses on spatial-temporal characteristics and influencing factors of G20 box office revenues. It uses some infrastructural and economic factors to evaluate the film geography. The subject is proportionately understudied in entertainment geography literature particularly focusing on the spatial distribution of the film industry and detecting temporal changes during the investigated period and has some merits from this point of view. However, the paper suffers from some dispersed and disorganized discussions in some sections. Moreover, some parts of the paper need to be revised thoroughly as they couldn’t cover what they should cover in their own sections. Therefore, some fundamental improvements are needed in the article to be publishable; the title, abstract, introduction, material and methods, analysis of influencing factors and conclusion and discussion sections have some shortages as follows.

 Title

It seems that the title doesn’t cover all the content of the article since it focuses mainly on China film industry while the title articulates the analysis is across the G20. Also, there are some analytical aspects of film geography in the paper that an international reader cannot understand it through the first glance to the title.

Abstract and Keywords

1.     This section is too general in terms of explaining the subject, research concerns and goals. It lengthens the results and lacks a complete scientific abstract standards.

2.     China film industry is written as a keyword while the G20 has noted as study area in the title

3.     Keywords don't cover all the research content.

Introduction

4.     This section fails to discuss the research concern robustly and remains disorganized as it brings various sub contents ranges from film economy to culture and space. I strongly suggest that the authors keep their focus on the main subject of the paper and deepen it the introduction so that readers can follow the main line of the research.

5.     The contributions of the research to international literature is not clear; what is the state of the art in terms of film geography theoretically and practically? What others have done so far in development of the subject methodically and in the case of intergovernmental forum such as G20? What are the most recent concerns of economic geography of film? I think these questions should be addresses for developing the article contributions

Research methods

6.     Study area needs to be more clarified via introducing map

7.     Data sources section is too long and has some redundant info (lines 80 to 87). Also this section should go to more details of data gathering method and sources of data

8. The research method is too general and one cannot follow the technical details of the methods and how they can be applicable in the analysis of the research variable, e.g. Markov chain, regression analysis

9. Selection of influencing factors section is somehow vague to me since it proposes some factors without documentation supports from other research. Moreover, justifications for the selected factors are rather confined to author’s interpretations and deprived from international state of the art literature.   

10. The quality of figures and maps presented in this section are too low so that most of them not readable

Conclusion and discussion 

11. The conclusion has brought up the first and then discussion has been written and the question is that how the findings can be concluded while discussions on the results remain untouched?

12. The analysis of the results should be expanded from China to the G20

 Language and Referencing

 13.  Referencing style should be corrected based on the Sustainability journal.

Round 2

Reviewer 1 Report

I have read the author's answers related to my comments, and the corrected version of manuscript, and I have noticed that the authors have taken in the consideration the majority of my comments. I do not have more comments.

Author Response

Reviewer 1 had no comments in round 2, so we didn't have any changes either.  We really appreciate reviewer’s  insightful comments in round 1. 

Reviewer 2 Report

The paper has been improved considerably. However, the are some comments that I think have been overlooked in the revision process.  

- Comment 7, part 2: Also this section should go to more detail of data gathering methods and sources of data

- Comment 10, the quality of maps has yet to improve so that a reader can identify the color codes
